# Does the SDMQ-9 Predict Changes in HbA1c Levels? An Ecuadorian Cohort

**DOI:** 10.3390/medicina58030380

**Published:** 2022-03-04

**Authors:** María José Farfán Bajaña, Jorge Moncayo-Rizzo, Geovanny Alvarado-Villa, Victor J. Avila-Quintero

**Affiliations:** 1Department of Health Sciences, Universidad de Especialidades Espíritu Santo, Guayaquil 090101, Ecuador; mariajosefarfanbajana@gmail.com (M.J.F.B.); jmoncayor@uees.edu.ec (J.M.-R.); 2School of Medicine, Yale University, New Haven, CT 06510, USA; victor.avila@yale.edu

**Keywords:** glycosylated hemoglobin, diabetes mellitus type 2, shared decision making, SDMQ-9

## Abstract

*Background and Objectives:* Diabetes mellitus affects 422 million people around the world, positioning it as a major health problem. According to the WHO(World Health Organization), 90% corresponds to type 2. The shared-decision making (SDM) is a method used to facilitate patient control, medication, maintenance, and assessment of health status according to their priorities and preferences. With the application of SDM in patients with diabetes, it is expected there will be an increase in treatment adherence and a reduction in HbA1c levels. The aim of this study is to determine the predictors of the change in HbA1c. *Material and Methods:* A sample of 76 participants attending as endocrinology outpatients was obtained. Data collected within the sample included: sex, age, educational level, body mass index, and the level of SDM using the SDMQ-9. In addition, HbA1c levels were measured twice: at baseline and three months after the first measurement. *Results:* The linear regression indicates that the level of SDM is a significant predictor of the change in HbA1c, specifically in men. However, the direction of the relationship was a somewhat opposite trend than we expected. Higher levels of SDM imply an increase in HbA1c rather than a reduction. *Conclusions*: Contrary to the literature, our results shows that elevated levels of perceived SDM may be associated with worse diabetic control. However, more investigation is needed as these results are not generalizable, due to the specific population used and the sample size. Furthermore, to better understand the effect of SDM on the change in HbA1c in patients with poorly controlled diabetes.

## 1. Introduction

About 422 million of people around the world have diabetes mellitus type 2 (DM2), according to the WHO [1]; in Ecuador, the prevalence of diabetes among people aged between 10–59, and 30–59 years is 3% and 10%, respectively [2] and the amount of people affected is on the rise [3]. The International Diabetes Federation foretells there will be 642 million individuals diagnosed with DM2 by 2040 [4]. In addition, chronic diseases such as DM2 are most often the cause of death and disability [5] despite available treatment options, due to a lack of adherence; in one study, only 20% of patients were considered to have good general adherence to diabetes treatment [6]. Increasing adherence may have a greater effect on health than improvements in specific medical therapy according to the WHO [7,8].

Thus, if emotional and physical factors intervene in medication adherence, shared decision making (SDM) could be a feasible method as it allows for patient-centered care and an improvement in the patient–physician relationship, in which both participate in the decision making process (concerns, cost-benefit, diagnostic preferences, monitorization and disease assessment are taken under consideration) using their personal perspective to improve treatment adherence and the course of the disease [9,10,11,12]. This method consists of the physician inviting the patient to participate in a dialogue, seeking to understand the condition, but also looking to present alternatives that meet therapeutic aims and enable patients to choose based on their preferences [11,12,13]. Better adherence will improve glycemic control and decrease health care resource utilization [6,14].

To understand physician–patient shared decision making and its influence on disease progress from the patients’ perspective, the 9-item Shared Decision Making Questionnaire (SDMQ-9) was developed [15]. This questionnaire evaluates the decisional process in medical encounters from both the patients (SDMQ-9) [15] and physicians’ perspectives (Doc-SDMQ-9) [16]. In addition, it has good acceptance, feasibility, and reliability. This tool has been validated in English [17], Spanish [18], and other languages [19,20,21].

The aim of this study is to determine the variables that predict changes in the levels of HBA1c, 3 months after attending their endocrinology medical appointment.

## 2. Materials and Methods

This study was carried out in the endocrinology outpatient department of “Teodoro Maldonado Carbo” Hospital of Specialties in Guayaquil, Ecuador from September to October 2017. This study protocol was examined and approved by the Committee of Ethics and Research in Human Beings (CEISH) of the Kennedy Hospital Group, authorized and evaluated by the Ministry of Public Health of Ecuador on 5 May 2016.

### 2.1. Sample

The participants involved in the study had the following criteria: being 18 years or older, diagnosed with DM2, with poor diabetic control defined by a level of glycosylated hemoglobin (HbA1c) > 7%; all had signed the informed consent form. The sample size was calculated with a 95% confidence interval and with 80% statistical power. The calculated sample was 107 participants.

### 2.2. Share Decision Making

To assess the level of shared decision making perceived by patients, the Shared Decision Making Questionnaire—9-items (SDM Q-9) was developed by Kriston et al. in 2018, in German [15]. This instrument is made up of nine questions with a response in a Likert-scale of six options (from “Totally disagree” to “Totally agree”). The scores of the questionnaire range from 0 to 45. To calculate the level of shared decision making, the score was multiplied by 20/9, so the level of SDM ranges from 0 to 100. The highest possible level of SDM was 100 and the lowest possible level of SDM was 0 [15]. For this study, the SDMQ-9 Spanish version was used, which was validated in an Ecuadorian population [18].

### 2.3. Procedure

All of the participants invited to the study had to sign the informed consent form, then sociodemographic data was collected (sex, age, level of education, private physician attendance, and BMI(Body mass index)). Moreover, the SDMQ-9 was applied to assess the level of shared decision making perceived by the patient.

Next, the participants were taken to a certified laboratory to collect the blood sample and measure the level of HbA1c. Then, two months and three weeks after the first blood sample, the participant was contacted by phone call. This second contact was to conduct a questionnaire about comorbidities, medication use, and diabetes control; also, the second blood sample collection was coordinated. Finally, the second blood sample was collected and analyzed by the certified laboratory three months after the first blood sample. This was done according to the NICE (The National Institute for Health and Care Excellence) guideline No. 28 from 2015 [22], which assessed that a three-to-six-month interval measurement is recommended. In addition, all of the samples were assessed by the same laboratory.

### 2.4. Statistical Analysis

The data were collected and analyzed using SPSS (Statistical Package for the Social Sciences)V.24. The qualitative variables were presented as percentages, while the quantitative data were presented with standard deviation. Associations were assessed with non-parametric tests as the variables had a non-normal distribution. The correlations between quantitative variables were assessed with the Spearman Rho test. Finally, a linear regression analysis was performed to identify the predictors of the change in HbA1c. The results are presented in tables and scatter-plots.

## 3. Results

There were 107 patients invited to participate in the study but only 89 patients agreed to participate (83.18%). However, 13 participants did not complete the questionnaires or did not attend the second blood sample collection. Therefore, the final sample was made up of 76 participants (response rate: 71.03%), expecting a statistical power at the end of 75%.

From the 13 participants who were excluded from the study, the majority were female (11/13) and had a mean age of 61.62 (SD: 8.45) with a range from 43 to 76. Regarding the body mass index (BMI), the majority were overweight and obese (11/13), with a mean value of 31.60 (SD: 8.75). The mean level of SDM was 50.60 (SD: 30.04; Min: 0; Max: 91.11). Finally, the mean level of HbA1c was 9.89 (SD: 1.77). The variables did not differ statistically significantly from the analyzed sample (*p* > 0.4).

The final sample was made up of 76 participants, where 57.9% were women. Some of the characteristics of the sample were already reported in another article [18]. Moreover, the majority of the participants were overweight or obese (84.2%) and only 18.4% of the participants attended private medical appointments. Table 1 presents the change in HbA1c according to the characteristics of the participants.

The mean level of SDM was 56.63 (SD: 27.27) with a range from 0 to 100. Moreover, the mean level of the baseline HbA1c was 9.72 (SD: 1.95) with a range from 7.10 to 17.10; the mean level of the HbA1c three months later was 9.51 (SD:1.86) with a range from 5.6 to 14.70. The change in the level of glycosylated hemoglobin (the difference between the HbA1c three months later and the basal HbA1c) was calculated. The mean difference of the HbA1c was −0.21 (SD: 1.53) with a range from −4.5 to 5.20.

No association were found between the difference in HBA1c and sex, level of education, age, BMI, and attendance at a private physician’s practice (See Table 1). Moreover, no correlation was identified between the difference in HbA1c and the level of SDM (r: 0.207 *p* = 0.072).

Linear regression analysis was performed to assess the predictors of the change in HbA1c. Sex, age, BMI, education level, attendance at a private physician’s practice, and level of SDM were the variables selected as the predictors on independent analysis. All variables, except the SDM level, were not significant to predict change in the HbA1c. The level of SDM significantly predicted the change in HbA1c (B: 0.013; CI: 0.001–0.026; *p* = 0.038) with a significant constant (B:−0.967; CI: −1.758–−0.177; *p* = 0.017). However, the explained variance by this model was 5.7% (R^2^ = 0.057) (see Figure 1). The equation that represents this relation was:ΔHbA1c=−0.967+0.013(SDM level)

Additionally, we performed a subgroup analysis. Prior to performing the regression, the distribution of the sex variable was tested with a non-parametric test. The distribution of females and males was homogenic (*p* = 0.207), indicating that a subgroup analysis was suitable. The regression analysis showed that, for men, the level of SDM statistically significantly predicts the change in HbA1c (B: 0.026; CI: 0.004–0.047; *p* = 0.02) with a significant constant (B:−1.46; CI: −2.811–−0.107; *p* = 0.035). This relation was determined by the equation ΔHbA1c=−1.46+0.026(SDM level), explaining 16.7% of the variance (R^2^ = 0.167). On the other hand, for females, this result was not significant (B:0.005; CI: −0.011–0.02; *p* = 0.53; R^2^ = 0.009) (see Figure 2).

## 4. Discussion

In diabetic patients, HbA1c levels are one of the most commonly used tests to assess the disease and to determine the prognosis [3,23]. Our results show poor diabetes control among the sample studied. Even though the mean HbA1c reduction was −0.21, the range of the change in HbA1c was from −4.5 to 5.2, which indicates an increase in their HbA1c.

Many studies have researched the variables that influence the changes in HbA1c levels; for example, diabetes duration, cholesterol levels [24], treatment adherence, and BMI reduction [25], among others. In addition, medication compliance could be affected by diabetes complications [26]; further, complications such as diabetic neuropathy [27] and microalbuminuria [28] involve a higher HbA1c and need more intense treatment. Moreover, other variables are taken into account such as self-rated health [29], family support, age [30], and the decision making process [31]. However, we found no association between the change in HbA1c and other variables such as age, sex, level of education, and BMI. Even though we did not find a correlation between the change in HbA1c and SDM levels (*p* = 0.072), the correlation direction was positive (*r* = 0.207). SDM levels increase following increases in HbA1c levels, indicating an opposite direction of correlation than we had expected.

The linear regression analysis showed that all of the variables except SDM were not significant in predicting the change in HbA1c. Moreover, in relation to education, in our study it was not correlated and did not predict the level of HbA1c in patients who have been diagnosed with diabetes, which concurs with Allen and McFarland’s findings [32]. Even though there is evidence of the relationship between BMI and diabetes [25,33], we found no association between the BMI and the HbA1c changes, as well as with age [30,34].

We found SDM levels to significantly predict the change in HbA1c (*p* = 0.038). In light of these findings, there is no evidence for shared decision making having a positive effect on reducing HbA1c, similar to the results of the DEBATE trial [31], an experimental trial where two groups of patients were compared. The experimental group were patients whose physician received training in patient-centered and shared decision making; and the control group were patients whose physicians did not receive the training. The results showed that there was no difference in the HbA1c levels between both groups [31].

In addition, the literature is inconsistent about the effect of sex on the levels of HbA1c. However, there is evidence showing males with diabetes having a higher risk for coronary complications [35,36]. The results from our study showed that, in men, the SDM level significantly predicts the change in HbA1c (see Figure 2), explaining 16.7% of the variance. Thus, in men with higher SDM levels, the HbA1c value will rise, indicating an increase in HbA1c levels relative to the baseline value This is the opposing approach to the concept of shared decision making, which involves taking into consideration the preferences and interests of the patients in order to increase treatment adherence based on a strong physician–patient relationship [9,10,12].

This study has several limitations. Sample size is an important limitation, for these reasons the results have to be interpreted cautiously. In addition, the number of HbA1c samples is a limitation. More relevant results may be obtained with a longer study taking blood samples every 3 months. Some variables that were not taken into consideration during the data collection were also limitations, such as the sex of the physician, the treatment that each patient was receiving, the final decision made during the consultation, etc.

## 5. Conclusions

To conclude, we found no correlation between changes in HbA1c and age, sex, education level, BMI, or attendance at a private physician’s practice. However, we found a correlation between the SDM level and a change in HbA1c. In addition, this was the only significant predictor of the change in HbA1c seen more strongly in men. Nevertheless, the results showed a somewhat opposite trend than we expected; indicating that with higher levels of SDM, HbA1c levels increase rather than reduce. However, our results are not generalizable due to our sample size and the specific Ecuadorian population selected. In order to better understand the effect of SDM on the change in HbA1c in patients with poorly controlled diabetes, more research is needed.

## Figures and Tables

**Figure 1 medicina-58-00380-f001:**
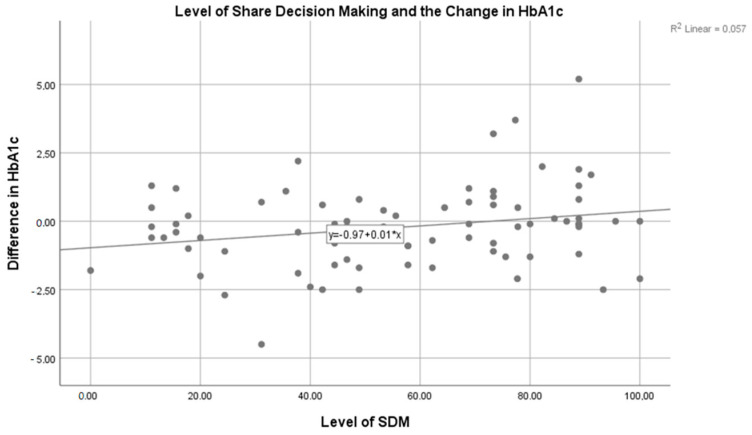
Relation of the level of SDM and the change in HbA1c. Scatter-plot assessing the regression model explaining the change in HbA1c with the Level of SDM by 5.7%. This model was statistically significant (*p* = 0.038).

**Figure 2 medicina-58-00380-f002:**
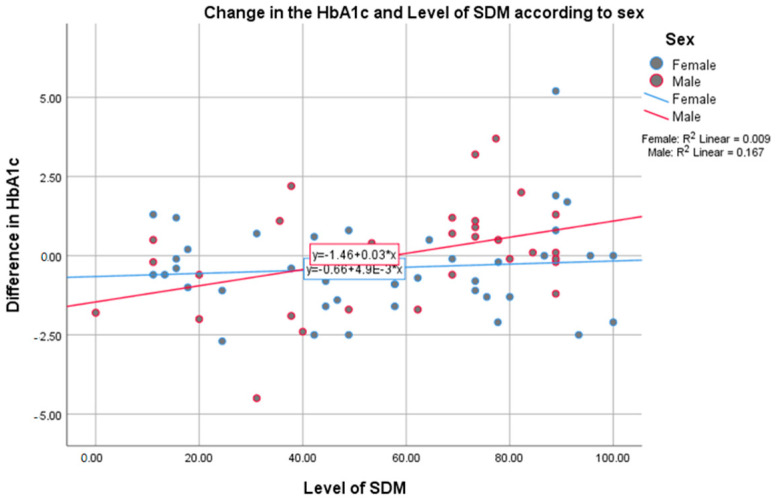
Change in the HBA1c and Level of SDM According to Sex. Scatter-plot assessing the regression models explaining the change in HbA1c with the level of SDM according to the sex. For males, the regression model significantly explained 16.7% of the variance (*p* = 0.02). On the other hand, for females, the regression model was not statistically significant (*p* = 0.53) and it explained only 0.9% of the variance.

**Table 1 medicina-58-00380-t001:** Demographic characteristics of the patients.

		Count (%)	SDM Level	Change in the HbA1c			*p* Value *
			Mean (SD)	Minimum	Maximum	Mean (SD)	
Sex	Female	44 (57.9)	55.66 (28.02)	−2.70	5.20	−0.38 (1.41)	0.101
	Male	32 (42.1)	57.97 (26.6)	−4.50	3.70	0.02 (1.66)	
Age Group	35–49 years old	10 (13.2)	58.89 (24.1)	−1.30	2.00	0.26 (1.15)	0.193
	50–64 years old	46 (60.5)	57.72 (27.79)	−4.50	5.20	−0.41 (1.65)	
	≥65 years old	20 (26.3)	53 (28.5)	−2.50	3.20	0.01 (1.36)	
Education Level	Without Studies	1 (1.3)	46.67 (.)	0.00	0.00	0.00 (.)	0.498
	Primary Education	19 (25)	53.33 (27.1)	−4.50	1.90	−0.25 (1.35)	
	Secondary Education	32 (42.1)	55.68 (27.1)	−2.70	3.70	−0.49 (1.45)	
	University Degree	24 (31.6)	60.93 (27.51)	−2.50	5.20	0.18 (1.75)	
Body Mass Index	Normal	12 (15.8)	42.22 (24.93)	−4.50	1.70	−0.75 (1.71)	0.383
	Overweight	31 (40.8)	60.27 (24.68)	−2.50	3.70	0.05 (1.43)	
	Obese	33 (43.4)	58.45 (29.4)	−2.50	5.20	−0.27 (1.53)	
Private Physician	Yes	14 (18.4)	53.17 (32.2)	−2.70	1.30	−0.58 (1.27)	0.503
	No	62 (81.6)	57.41 (26.27)	−4.50	5.20	−0.13 (1.58)	

* Relation of the change of HbA1c in the variables.

## Data Availability

The datasets used and/or analyzed during the current study are available from the corresponding author on reasonable request.

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
