# Peer review of "Does the SDMQ-9 Predict Changes in HbA1c Levels? An Ecuadorian Cohort"

_medicina, 2022, doi:10.3390/medicina58030380_

Round 1
Reviewer 1 Report
The article analysis the impact of SDm on glycemic control. The results and discussion chapter must be improved. Please analise the impact of diabetes complications on disease management. Please check: Diabetic neuropathy: A narrative review of risk factors, classification, screening and current pathogenic treatment options (Review). Exp Ther Med 22: 690, 2021 and Prediction Models of Albumin Renal Excretion in Type 2 Diabetes Mellitus Patients. Rev. Chim. 2019 Nov;70(11):3802-3807. Available from: https://doi.org/10.37358/RC.70.19.11.7650
Author Response
We appreciate your suggestion. Thank you for your review. The discussion were considered for its improval. The information and the bibliography was added to the document as you suggested. (Line 167-170)
Reviewer 2 Report
Review of: Does SDM-Q9 is able to predict changes in HbA1c levels? – an Ecuadorian cohort.
This manuscript testing the effectiveness of SDM-Q9 as a predictor to change in control of diabetes by the patients as indicated by the levels of HbA1c. There is a major need to test and evaluate the effectiveness of SDM-Q9 questioner on clinical outcomes. Analysis presented here builds on previous study confirming the validity of SDM-Q9 in Ecuadorian population and here correlating SDM scores with HbA1c as indicative for controlled diabetes. However, some concerns remain regarding the extent of the statistical analysis, limitation of the conclusions and explanation to other sources of variability that can explain the results.
Major comments:
- Data from this study was published by the authors in 2019 (ref [18]) to support the validity of SDM-Q6 questioner in Spanish translation to patients in Ecuador. Results of HbA1c were not published in 2019 and remain unique to the current 2022 manuscript. While it is understandable why authors first published the questioner validation and in earlier time point, the reuse of same study design and same patient population impairs the novelty of current report.
- Lines 20-23; Different populations, in different countries, will respond differently to SDM-Q9-based medical decision making. Therefore, any conclusion should mention the results represent specific Ecuadorian cohort / population of patients. Not mentioning the population implies a stronger conclusion that was not tested (e.g., applicability of conclusion to other non-Ecuadorian populations).
- Why the 3-month time point was chosen for a second blood test (e.g., Line 58)?
- Lines 91-93, Did both the first and second blood samples of each patient were analyzed in the same laboratory (e.g., excluding between-site variability)? This should be stated to the reader.
- Figures 1 and 2. In addition to a title, it is customary that each figure will have legend with short description of the result, technical information (e.g., report in figure’s legends the number of subjects in each group, statistical method, p-value etc.).
- Why there was no outlier exclusions procedure on data from patients with very unusual HbA1c levels or with unusual calculated difference in HbA1c levels?
- Was subgroup analysis performed on addition variables such as age (by ranges), education level and private physician?
- Did authors conduct analysis of covariance? (e.g., accounting for age or BMI for example, can explain some of the variance observed in Figure 1; interaction effects between these variables are also possible).
- The levels of SDM score suffer from high variability (e.g., Line 123, SD of 27 while the mean score is 56); The variability source was partially explained by Sex. (A) When subgrouping by sex what was the mean and SD of SDM score for the male and female groups? (B) Mean and SD of SDM score can be added to all groups in Table 1 and may provide straightforward and valuable information to explain possible sources variability.
- Discussion can be improved by providing information on some of the limitations of the Ecuadorian population in the study area that may influenced patient adherence to treatment (e.g., availability and access medical treatment and medicine, cost of medicine relative to income etc.)
Minor
- Line 104, “83,18%” into “83.18%” to be consistent.
- Line 140, superscript the two (R2) in R-square.
- Few of the reference are articles not in the English language.
Author Response
Thank you for your comments. Please find our response to your comments in the following paragraphs.
Major comments:
- Data from this study was published by the authors in 2019 (ref [18]) to support the validity of SDM-Q6 questioner in Spanish translation to patients in Ecuador. Results of HbA1c were not published in 2019 and remain unique to the current 2022 manuscript. While it is understandable why authors first published the questioner validation and in earlier time point, the reuse of same study design and same patient population impairs the novelty of current report.
Your comment is greatly appreciated. Our goal was to increase the budget and get more patients after validating the questionnaire, but that was not feasible in 2019. Furthermore, the pandemic of COVID-19 in 2020 and 2021 complicated the scenario, which is why we decided to complete the article this year to minimize wasting more time in the future.
- Lines 20-23; Different populations, in different countries, will respond differently to SDM-Q9-based medical decision making. Therefore, any conclusion should mention the results represent specific Ecuadorian cohort / population of patients. Not mentioning the population implies a stronger conclusion that was not tested (e.g., applicability of conclusion to other non-Ecuadorian populations).
Thank you for your observation. The information was added in the abstract (line 21-24) and the conclusion section of the article (line 208, 209)
- Why the 3-month time point was chosen for a second blood test (e.g., Line 58)?
As our study was conducted in poorly controlled diabetic patients, losses are possible and even bigger when the study timeframe increases1. Some authors consider changes in hemoglobin may be noticeable in 3 months because it is the normal time for the life cycle of the hemoglobin. Also, the NICE guideline No. 28 from 2015, which assess that a three-to-six-month interval measures are recommended2. The study protocol was made in 2016, so we took this guideline into account. We added this information to the manuscript (line 94-96).
- Dettori JR. Loss to follow-up. Evid Based Spine Care J. 2011;2(1):7-10. doi:10.1055/s-0030-1267080.
- Blood glucose management [Internet]. Type 2 diabetes in adults: management. National Institute for Health and Care Excellence (UK); 2015 [cited 2022 Feb 16]. Available from: https://www.ncbi.nlm.nih.gov/books/NBK553510/
- Lines 91-93, Did both the first and second blood samples of each patient were analyzed in the same laboratory (e.g., excluding between-site variability)? This should be stated to the reader.
Yes, the samples were analyzed in the same laboratory, avoiding between-site variability. This was added to the manuscript (line 96-97).
- Figures 1 and 2. In addition to a title, it is customary that each figure will have legend with short description of the result, technical information (e.g., report in figure’s legends the number of subjects in each group, statistical method, p-value etc.).
The request information was added
- Why there was no outlier exclusions procedure on data from patients with very unusual HbA1c levels or with unusual calculated difference in HbA1c levels?
Two analyses were performed to known if outliers may influence the model results. The Cook’s distance of the observations of the regression model showed the maximum value was 0.189. This value was from an extreme observation data (Diff. of HbA1c = 5.2). This is interpreted as there is no significant influence of those extreme values on the model1. Moreover, Mahalanobis’ distance for the data of the regression model was not significant (p=0.048). Typically, a p-value less than 0.001 is considered to be an outlier1. According to these data, no outlier exclusion was performed.
- Stevens, J. P. (1984). Outliers and influential data points in regression analysis. Psychological Bulletin, 95(2), 334–344. https://doi.org/10.1037/0033-2909.95.2.334
- Was subgroup analysis performed on addition variables such as age (by ranges), education level and private physician?
No other subgroup analysis was performed due to a heterogenic distribution of the variable (p-value for distribution p<0.05). The only variable which was homogenic distribute was sex (p=0.207) (line 149-151).
- Did authors conduct analysis of covariance? (e.g., accounting for age or BMI for example, can explain some of the variance observed in Figure 1; interaction effects between these variables are also possible).
Covariances were analyzed. A multiple linear regression was performed for all the variables that may influence in the model by a stepwise method. Therefore, the explained variables presented were the only significant. No other variable showed influence.
- The levels of SDM score suffer from high variability (e.g., Line 123, SD of 27 while the mean score is 56); The variability source was partially explained by Sex. (A) When subgrouping by sex what was the mean and SD of SDM score for the male and female groups? (B) Mean and SD of SDM score can be added to all groups in Table 1 and may provide straightforward and valuable information to explain possible sources variability.
The request information was added.
- Discussion can be improved by providing information on some of the limitations of the Ecuadorian population in the study area that may influenced patient adherence to treatment (e.g., availability and access medical treatment and medicine, cost of medicine relative to income etc.)
Since the medication and medical appointments are being covered by the government’s social insurance, this won’t make a significant difference in their adherence to treatment.
Minor
- Line 104, “83,18%” into “83.18%” to be consistent.
- Line 140, superscript the two (R2) in R-square.
- Few of the reference are articles not in the English language.
The minor observations were corrected.
Round 2
Reviewer 2 Report
I would like to sincerely thank the authors for their responses and for addressing my comments. In the revised version (v2), all of my comments were adequately addressed and the authors added the appropriate information to the manuscript. Much success!